# Community Consensus Guidelines to Support FAIR Data Standards in Clinical Research Studies in Primary Mitochondrial Disease

*Amel Karaa,\* Laura E. MacMullen,\* John C. Campbell, John Christodoulou, Bruce H. Cohen, Thomas Klopstock, Yasutoshi Koga, Costanza Lamperti, Rob van Maanen, Robert McFarland, Sumit Parikh, Shamima Rahman, Fernando Scaglia, Alexander V. Sherman, Philip Yeske, and Marni J. Falk\**

Primary mitochondrial diseases (PMD) are genetic disorders with extensive clinical and molecular heterogeneity where therapeutic development efforts have faced multiple challenges. Clinical trial design, outcome measure selection, lack of reliable biomarkers, and deficiencies in long-term natural history data sets remain substantial challenges in the increasingly active PMD therapeutic development space. Developing "FAIR" (findable, accessible, interoperable, reusable) data standards to make data sharable and building a more transparent community data sharing paradigm to access clinical research metadata are the first steps to address these challenges. This collaborative community effort describes the current landscape of PMD clinical research data resources available for sharing, obstacles, and opportunities, including ways to incentivize and encourage data sharing among diverse stakeholders. This work highlights the importance of, and challenges to, developing a unified system that enables clinical research structured data sharing and supports harmonized data deposition standards across clinical consortia and research groups. The goal of these efforts is to improve the efficiency and effectiveness of drug development and improve understanding of the natural history of PMD. This initiative aims to maximize the benefit for PMD patients, research, industry, and other stakeholders while acknowledging challenges related to differing needs and international policies on data privacy, security, management, and oversight.

## 1. Introduction

Primary mitochondrial diseases (PMD) are genetic disorders known for their extensive clinical and molecular heterogeneity,[1–4] whose etiology is now recognized to result from pathogenic variants in any of more than 350 genes across both nuclear and mitochondrial DNA genomes.[5,6] PMD are rare diseases, and hundreds are ultra-rare. PMD may impair the function of any organ system at any age, leading to chronic, complex, and progressively disabling conditions.[2,7] Therapeutic development efforts for PMD face multiple challenges due to this phenotypic complexity, lack of predicable phenotype–genotype correlation, general lack of validated outcome measures or reliable biomarkers,[8] a wide spectrum of morbidity and mortality (with limited natural history studies), and fluctuating, episodic symptoms.[9]

Several PMD therapies sponsored by the life sciences industry have been trialed since 2010. However, most of these

A. Karaa
Department of Genetics, Massachusetts General Hospital
Harvard Medical School
Boston, MA 02114, USA
E-mail: akaraa@mgh.harvard.edu

L. E. MacMullen, M. J. Falk
Mitochondrial Medicine Frontier Program, Division of Human Genetics, Department of Pediatrics
Children's Hospital of Philadelphia
Philadelphia, PA 19104, USA
E-mail: dennisl@chop.edu; falkm@chop.edu

J. C. Campbell
Minovia Therapeutics
Cambridge, MA 02138, USA

J. Christodoulou
Murdoch Children's Research Institute and Department of Paediatrics
University of Melbourne
Melbourne, Victoria 3052, Australia

PMD clinical trials have failed to reach their primary endpoint. Contributory factors include problems with patient selection, limited consideration of molecular etiologies, and use of outcome measures not specific for or validated in PMD.[10] In 2015,

B. H. Cohen
Department of Pediatrics and the Rebecca D. Considine Research Institute
Akron Children's Hospital
Akron, OH 44308, USA

T. Klopstock
Friedrich-Baur Institute, Department of Neurology
University Hospital
LMU, Munich 80336, Germany

T. Klopstock
German Center for Neurodegenerative Diseases (DZNE)
Munich 80336, Germany

T. Klopstock
Munich Cluster for Systems Neurology (SyNergy)
Munich 80336, Germany

T. Klopstock
German Network for Mitochondrial Disorders (mitoNET)
Munich 80336, Germany

Y. Koga
Department of Pediatrics and Child Health
Kurume University School of Medicine
Kurume 830-0011, Japan

C. Lamperti
UO Genetics and Neurogenetics
Fondazione IRCCS Instituto Neurologico C. Besta
Milan 20126, Italy

R. van Maanen
Khondrion BV.
Nijmegen 6525 EX, The Netherlands

R. McFarland
WCMR Newcastle University
Newcastle NE1 7RU, UK

S. Parikh
Cleveland Clinic Foundation
Cleveland, OH 44195, USA

S. Rahman
UCL Great Ormond Street Institute of Child Health and Great Ormond Street Hospital for Children NHS Foundation Trust
London WC1N 1EH, UK

F. Scaglia
Department of Molecular and Human Genetics
Baylor College of Medicine
Houston, TX 77030, USA

F. Scaglia
Texas Children's Hospital
Houston, TX 77030, USA

F. Scaglia
Joint BCM-CUHK Center of Medical Genetics
Prince of Wales Hospital
Hong Kong SAR NT, China

A. V. Sherman
Department of Neurology, Massachusetts General Hospital
Harvard Medical School
Boston, MA 02114, USA

P. Yeske
United Mitochondrial Disease Foundation
Pittsburgh, PA 15239, USA

M. J. Falk
Department of Pediatrics
University of Pennsylvania Perelman School of Medicine
Philadelphia, PA 19104, USA

the mitochondrial disease clinical trials working group organized a Critical Path Innovation Meeting (CPIM) with the Food and Drug Administration (FDA) to discuss general approaches to overcome the challenges to developing therapies for PMD, focusing on clinical trial design, outcome measures, biomarkers, and regulatory pathway considerations.[11] Life sciences industry investment in drug trials for PMD is growing. However, limited patient numbers, limited disease-specific natural history data, phenotypic complexity, inconsistent clinical trial eligibility criteria, and differential responses to treatments between individual patients call for a concerted effort to optimize the clinical trial design.

When asked to evaluate the merit of a clinical trial protocol and outcome measure selection, industry, academia, and regulatory agencies most often report deficiencies in long-term natural history data sets for PMD as one of the most limiting factors in optimal clinical trial design. Even when a collection of natural history data exists, they typically either lack the necessary longitudinal component or are retrospective data sets, collected in a unique institutional format based upon clinic visit measurements with varying and often incomplete assessments. Common outcome measure data are required to appreciate natural disease progression across a population. Industry also faces the obstacle of challenging and expensive access to institutional databases with varied patient privacy restrictions and institutional policies supporting a system of incentives surrounding access to proprietary databases. In addition, the patient privacy regulations can vary between institutions and countries. These deficiencies and challenges collectively contribute to prolonged and burdensome interventional and noninterventional clinical trials that are collecting data in an attempt to address background knowledge gaps that may have been informed earlier in the process with the availability of more complete, longitudinal, and robust patient data.

Other major challenges in the increasingly active PMD therapeutic development space are the high risk, expense, and multidisciplinary expertise required to design trials, which require a range of ongoing stakeholder input and collaboration. Greater data transparency among stakeholders in PMD clinical research is needed to inventory and harmonize data existing in silos across international academic research clinical networks, industry-sponsored pre-trial registries, and clinical trials. Such transparency is crucial for both sharing of data, which includes patient-centric variables from clinical and patient-reported measurements (e.g., genotypes, laboratory values, patient diaries, questionnaires), as well as metadata entities that come from ontologies, vocabularies, and schemas (e.g., accession codes, licenses, table headings, disease names, gene names, chemical entities, units). As a crucial precursor to increased data transparency and community-wide data sharing, data must first be made sharable in a FAIR (findable, accessible, interoperable, reusable) structure format in accordance with the FAIR data principles: findability, accessibility, interoperability, and reusability.[12] Establishing a consistent framework for compiling individual subject data and sharing metadata with appropriate data protections in place could enable PMD research by facilitating meta-analyses and assuring research results from one study were validated and reproducible across other studies. Increased data transparency and widespread sharing of research study data also are likely to: 1) improve the efficiency of future

studies design, 2) increase the probability of success and acceptance, thereby encouraging and accelerating trials of candidate therapies in PMD, optimizing patient access to new effective therapies, maximizing limited resources and minimizing subject risks, and 3) reduce duplication in interventional and noninterventional clinical trials, to the ultimate benefit of people with PMD who participate in clinical research.

Data harmonization efforts across patient registries are underway in several rare disease groups and may streamline therapeutic efforts. Historically, progress of these efforts has been hindered by siloed and uncoordinated datasets,[13] though recent initiatives to harmonize datasets using internationally accepted phenotype terminology have gained some traction. The extensive phenotypic and genotypic heterogeneity of PMD requires researchers to determine whether subgroups can be identified on the basis of specific disease definitions or therapeutic targets and whether clinical research data can reliably be extrapolated from one ultra-rare disease subtype to inform likely outcomes in another subtype, or even inform the management of more common diseases with a component of secondary mitochondrial dysfunction. The International Classification of Inherited Metabolic Disorders (ICIMD) is a global initiative working to address these gaps through harmonization of mitochondrial disease classification.[14] Ultimately, harmonization and community sharing of clinical research study metadata are of critical importance in PMD therapeutic development. These require the establishment and acceptance of data capture standards to be adopted by clinical consortia and research groups to eventual enablement of data sharing and distribution across compatible platforms. This manuscript is the initial step for the international PMD community to acknowledge current gaps that exist in data collection and data sharing and declares the imperative to establish a unified way forward to establish proper standards and protocols for FAIR Data Standards of PMD clinical and research data.

## 2. Current Landscape for PMD Clinical Research Data Sharing: Obstacles and Opportunities

In the current environment, there is a bias toward publishing only significant research findings that meet the research objective. The same is true for research findings presented at scientific meetings. However, reviewing and evaluating data from unsuccessful trials would also be informative for planning future clinical trials and improving understanding of and addressing contributing factors to trial failure, other than lack of drug efficacy. Published information includes partial data points with content directed by scientific journal guidelines, required by ClinicalTrials.gov, and levied by the competitive nature of industry research with selective data sharing to protect proprietary knowledge and commercial advantage. A large amount of data are also stored under confidentiality and nondisclosure agreements for Pharma studies and/or by grant-funded studies by academic researchers, pharmaceutical companies, and government agencies such as the FDA, European Medicines Agency (EMA), and the National Institutes of Health, with missed opportunities for further exploration and advancing scientific knowledge.

Efforts toward the standardization of clinical trials' data do exist, as drug developers are required to conform to the standards of the Clinical Data Interchange Standards Consortium (CDISC)[15] prior to submitting data to authorities. Granted, such data conversion is performed on successful trials only. However, little standardization or harmonization exists among clinical research natural history data, or between observational clinical research and interventional trial protocols. Absence of professional data management efforts contributes to deficits of high-quality, data sets, as often data missingness and data entry errors prevent accurate comparisons or even the possibility of harmonization across multiple research studies. Industry studies generally do not specify the methods of data generation with cross-studies' data integration in mind, which could allow for consistent data collection. Identification of these gaps may be expensive and time consuming to discern. Nonetheless, the accessibility of pooled data for current clinical trials in PMD would serve as a crucial starting place for data harmonization, allowing researchers to combine and compare outcomes from clinical trials in PMD with similar enrollment criteria or investigating drugs with similar mechanisms of action. Collection of these data will require a collaborative effort with streamlined incentives for all PMD community stakeholders.[16]

Clinical research data sharing offers an opportunity to heighten public trust in clinical trials by making the clinical research process more transparent. A scheme to share data in a consistent way would encompass an operating platform that could evaluate and index the diverse variety of available data into an aggregated, pooled, global catalog of data content. This, in turn, will allow data objects from any clinical research activity to be more easily retrieved and reported in detail to potential users, leading to secondary analyses studies, thus expanding knowledge generation. This could improve public health by enhancing patient safety, improving best clinical practices, and increasing the probability of success in drug development. As interest and awareness in PMD therapeutic development grow, the generation of an expanded dataset may be useful beyond the single study being considered, across the broader stakeholder spectrum.

The PMD patient community would likely find greater motivation to participate in clinical research studies if it felt confident that all research subject data points would be maximally utilized to apply learnings to other PMD clinical trials, as well as to more common diseases involving secondary mitochondrial dysfunction. Clinical trial participant attitudes toward the return of research results support this conclusion, as many cite clinical significance and understanding of disease as their reason for wanting to receive individual and aggregated study results.[17] Indeed, a frequent complaint raised by clinical trial participants is a lack of access to these study results, leaving participants uncertain of their contribution to the overall research effort and the advancement of knowledge in the field. As the patient perspective is critically important when considering clinical trial and/or medical record data sharing,[18] patients and families must be recognized as key stakeholders in data generation and sharing. Additionally, shared decision-making between patients, healthcare providers, and drug developers is crucial for designing relevant and impactful clinical trials and for maximizing the impact of research in rare diseases.[19] Patient engagement allows researchers to understand the scope of disease burden, incorporate patient preferences for symptom-targeted treatment, and recognize and address barriers to participation in clinical trials.[20]

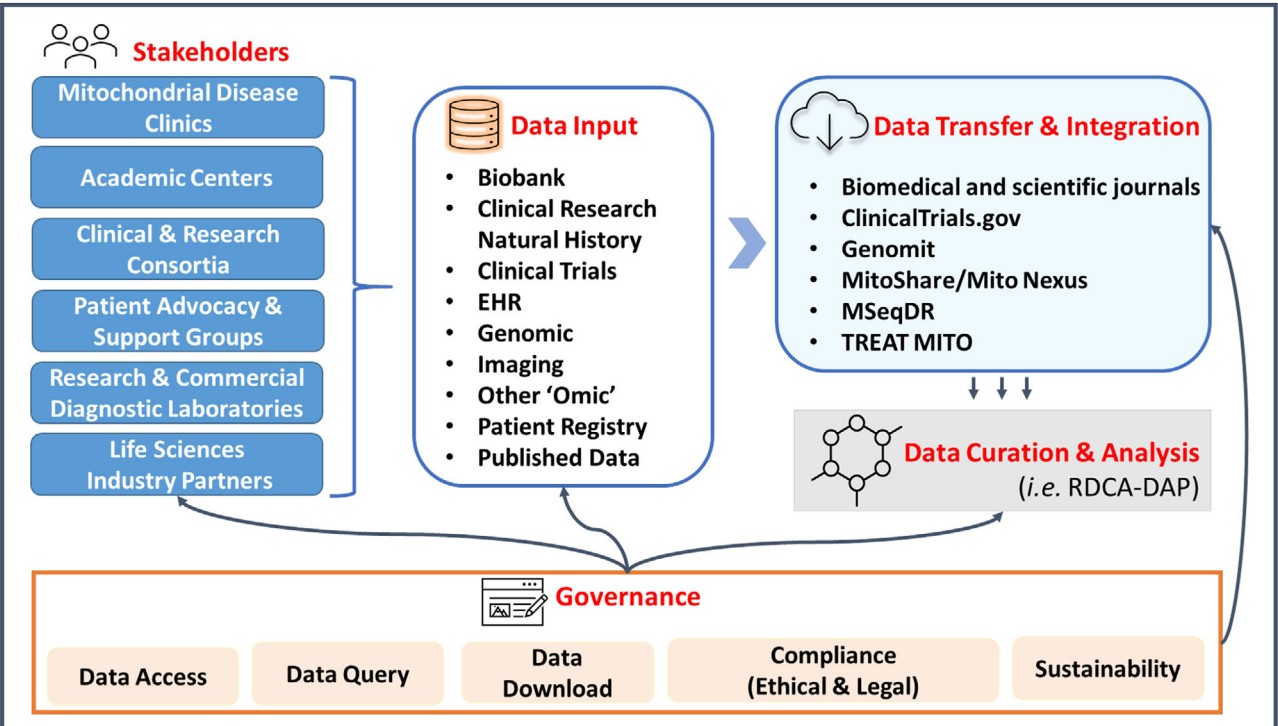

**Figure 1.** Current landscape for PMD clinical research and data sharing. EHR, electronic health records; MSeqDR, mitochondrial disease sequence data resource consortium; TREAT MITO, translational REsearch advancing therapy in MITOchondrial diseases; RDCA-DAP, rare disease cures accelerator-data and analytics platform.

During the 5th Wellcome Genome Mitochondrial Medicine Meeting held in late 2020, which was focused on therapeutic development in primary mitochondrial disease, extensive discussions were held among members of the international PMD community during sessions on clinical trial updates and challenges. These discussions highlighted the fundamental need for PMD community data standards, unified data platforms, and protocols to facilitate the adoption of effective data sharing across interested stakeholders, including industry partners, academic institutions, clinicians, and patient advocates involved in the drug development process (**Figure 1**). However, several concerns and challenges were raised, including the expense of initiating and maintaining this endeavor, funding for data collection and data entry, data ownership, various international data protection /privacy regulations, access, licensing, and contractual concerns, "FAIR" data resource creation and sharing, and long-term capacity building. Furthermore, there are concerns regarding data governance, ensuring that data access requests are thoroughly reviewed and only granted for scientifically valid research purposes. Ultimately, community data sharing can be accomplished only if the legitimate concerns of all stakeholders involved in the process are adequately addressed, including compliance with changing privacy laws (e.g., evolving protections of genetic data), competition, costs, funding challenges, incentives, and willingness to collaborate. Clear and uniformly applied guidelines must exist, including the development of uniform content for informed consent forms, consistent language for sharing, and agreements for standard data use requests and material data transfer. Additionally, the research community must implement strategies for secure integration of data originating from the same research subjects participating in multiple studies, through the utilization of disease-specific Global Unique Identifiers (GUIDs) and their derivatives. Compliance with data standards should be monitored by the community through a fair and transparent process that incentivizes collaboration and discourages noncompliance.

Appropriate safeguards must be in place to protect pharma companies willing to share clinical research data from individuals, institutions, or companies seeking to exploit shared data to commercial, competitive, or litigious advantage. Indeed, industry has a responsibility to shareholders, investors, employees, and research subjects to respect their significant investment of time and resources required to develop potential therapies for PMD. As the PMD therapeutic development community is relatively small, researchers and drug developers are typically familiar with the broader scope of research efforts ongoing in the PMD field. Sharing details of trial design, target population, and study endpoints can potentially expose proprietary information, thereby compromising a company's investment and disincentivizing clinical trial sponsors. These challenges can likely be overcome with careful planning, ongoing communication, and standardized processes, including consideration of appropriate timing for community data sharing after the completion of primary data analysis and interpretation. A stepwise approach might be necessary, starting with the sharing of noninterventional trial data and interventional trial data collected prior to the administration of study treatment. Certainly, pooling control-arm data from controlled clinical trials is paramount. Access to data generated in clinical trials is

**2100047 (4 of 11)**

frequently restricted, not only to avoid exposing confidential commercial information but also to protect participants' privacy and clinical trial integrity by avoiding the introduction of bias during interpretation. Researchers seeking to share trial data with the community for meta or secondary analyses may need to engage data scientists and develop consistent anonymization processes, data storage, and sharing standards to optimize data security and prevent potential data breaches.

Clinical research data are generally subject to multiple global regulatory standards. Typical requirements include removal of identifying information, either through complete data anonymization or by assigning a unique code to patient data (which may be particularly challenging when considering genetic data in ultra-rare disease) as well as data aggregation to comply with national data regulations. Institutional, corporate, international, regulatory, and legal guidelines already exist, and they need to be thoroughly understood in relation to each other before data sharing can be appropriately implemented. General Data Protection Regulation in the European Union (EU GDPR) should be considered for any data export outside the EU. For example, a recent Court of Justice of the European Union (Schrems II on July 16th, 2020) declared "the European Commission's Privacy Shield Decision invalid on account of invasive US surveillance programs, thereby making transfers of personal data on the basis of the Privacy Shield Decision illegal [...] and stipulated stricter requirements for the transfer of personal data based on standard contract clauses."[21] Although clinical research data may be exempt from this decision as stipulated in GDPR. Other rare disease groups have demonstrated ways in which clinical research data sharing could be successfully achieved, including Amyotrophic Lateral Sclerosis Consortium (PRO-ACT, NEALS, Project MinE),[22,23] the Canadian Neuromuscular Disease Registry (CNDR),[24] and the GNE Myopathy Disease Monitoring Program (GNEMDMP).[25] Furthermore, groups such as the Global Alliance for Genomics Health (GA4GH) have launched efforts to facilitate data sharing through a Data Use Ontology[26] for semantically tagging datasets to improve findability and digital passports for authenticating user identity to streamline access to genomic datasets.[27]

The PMD community should seek to advance discussions on overcoming data sharing challenges with all stakeholders, including research participants, study sponsors, regulators, investigators, research institutions, journals, professional societies, and data scientists. These collaborative discussions are necessary to achieve the goal of community sharing in acceptable formats using data standards from PMD clinical research studies including natural history study and clinical intervention trials.

## 3. Proposed Formats and Methods for Sharing PMD Clinical Research Data

The growing momentum that exists within the PMD community to pursue robust clinical research has reached the stage where the expected norm should be to foster responsible and comprehensive data sharing. A first step in this process is to define and identify the data attributes that need to be captured and standardized using structured data formats.

### 3.1. Metadata for PMD Clinical Research Studies

Metadata are structured data that describe data from individual studies. "Pervasive in information systems," metadata "come in many forms and are key to the functionality of systems holding data content, enabling users to find items of interest, record essential information about them, and share that information with others."[28] Three general types of metadata include 1) administrative (technical, preservation, rights), 2) descriptive (resources, content, characteristics), and 3) structural (how objects exist in relationship to each other).[29] For metadata to be useful and informative, the creation of a PMD community-approved list of the metadata attributes that can and should be captured as well as a content standard are needed, as well as a quality control process to maintain the accuracy, completeness, consistency, and interoperability of the metadata.[30] For PMD, metadata describing any and all properly consented, and privacy-protected individual data aggregated from clinical research studies may be useful. Demographics, safety labs, concomitant medications, medical histories, vital signs, efficacy parameters, and adverse events data can be standardized to establish data dictionaries or broaden the use of existing data standards like CDISC. Outcomes and established biomarkers may be "mixed" with experimental biomarkers and other more innovative, future data collections. With the support of a data manager, more granular data may also be incorporated including clinical data (physician and patient reported), histologic and/or biochemical data (e.g., assayed in blood, urine, cerebrospinal fluid, tissue biopsies, cell lines), imaging data, electrophysiology data, and multiomics data (genetic and genomic data, transcriptomic data, proteomic data, lipidomic data, epigenomic data, and metabolomic data). This information will improve endpoint selection, help predict treatment response, and support study sample size selection.

### 3.2. Comprehensive Data Sets and Centralized Data Sharing for PMD Clinical Research Data

Large clinical data sets may be used as calibration tools for identifying statistically significant and biologically meaningful insights in populations. Furthermore, they are of particular value for studying rare diseases. Wide sharing of detailed data from each research study beyond the single study publication of positive results will be beneficial to support genotype–phenotype correlations, inform sample size calculations, and study recruitment strategies, and may lead to improved measures of efficacy or non-efficacy in future studies through a selection of the most appropriate disease population. Large clinical data sets can also support the design of more effective PMD therapy trials by helping to identify and validate relevant and meaningful outcome measures and harmonize clinical assessment and case report forms. Such data sets may also aid in improving PMD clinical care standards, as they will serve as a source of more information than the limited data sets that are currently published or otherwise inaccessible to key stakeholders.

Sharing large clinical datasets with the broader scientific community can potentially be accomplished in many ways, including in journals as supplemental files at the time of clinical study publication. However, developing a community platform to facilitate

**2100047 (5 of 11)**

sharing harmonized and anonymized data from clinical research studies in an open-access clinical database would support and likely drive global innovation in clinical research and therapeutic development (e.g., PRO-ACT[31]). While centralization of data in a single location may not be required, knowing where and how the data are accessible across the clinical research ecosystem of PMD is paramount. Cataloging clinical research study type is important, delineating what information was obtained in natural history studies or clinical intervention trials. Connecting clinical and research data to clearly delineated genetic data, biobank repositories, imaging biobanks, multiomics datasets, and patient-reported outcomes using the same defined language ontology is essential.

Establishing and optimizing data sharing standards and structures for PMD clinical research studies will be an iterative and staged process. A natural starting point beyond data publicly available on ClinicalTrials.gov may be to begin sharing nonproprietary placebo group data, baseline data, and/or data from preintervention run-in studies. The challenge will be to share sufficient metadata to make sense of the available clinical trial data without compromising the trial's integrity or results publication, or regulatory proceedings while respecting the trial sponsor's responsibility to protect intellectual property and company investment in the therapy under investigation. Key challenges to consider here, include deciding on what metadata to share, how to get everyone on board for sharing their metadata, and under which protocol and governance. Ultimately, finding a path and appropriate timing for deposition and sharing of both placebo control and intervention arm data from clinical trials will be valuable. Knowing the precise study context and precise method with which data selection and collection was undertaken will be paramount to subsequent data analyses. Again, a key element of success will be the diligent and fair addressing of legitimate concerns unique to individual stakeholders in the process and including the patients that will ultimately provide the data.

### 3.3. Methods for PMD Clinical Research Data Sharing

Building robust data sharing standards and platforms has been a priority of many research and healthcare organizations worldwide for several years. The benefits of data sharing have been fully endorsed by the U.S. National Academies of Science, Engineering and Medicine, the Innovative Medicines Initiative (IMI) in Europe, global patient advocacy groups, the International Committee of Medical Journal Editors (ICMJE), government agencies, the National Institutes of Health (NIH), and the Coalition for Accelerating Standards and Therapies.[32] Efforts to strengthen data sharing policies and improve data standards have led to recommendations by FORCE11 to follow the FAIR principles[12,33] with emphasis on applying robust data interoperability. Attempting to share nonstandard data formats would make an aggregation of datasets from different sources considerably more time consuming and costly, compromise data quality and integrity, and lead to interpretation errors that would limit the accuracy, accessibility, and reusability of data.[34]

Use of NIH developed Common Data Elements (CDE), close to the International Organization for Standardization (ISO) and International Electrotechnical Commission (IEC) 11179-3 Metadata

registry model and basic attributes (ISO/IEC 11179) standard[35] was initially considered to be the solution for data sharing. However, this standard has since proved to be poorly suited for global use, due to its being insufficiently specific for certain research areas and difficult to deploy easily and rapidly.[34] Several other models of metadata sharing are being developed for rare disorders, and each platform has benefits and limitations. The United Mitochondrial Disease Foundation (UMDF) patient registry, mitoSHARE, uses JSON-format, which has a simple design and flexibility that makes it easy to read, understand, and manipulate in a range of programming languages, and is ubiquitous for sending data between Web servers, browsers, and mobile applications.[36] The mitoSHARE format is extensible and allows for linking to other known industry standards.

The Clinical Data Interchange Standards Consortium (CDISC), supported by the National Cancer Institute (NCI) Cancer Data Standards Registry and Repository (caDSR) and Enterprise Vocabulary Services (EVS) group,[15] is a global standards development organization that follows an ISO-recognized process and has been globally accepted as an adequate tool for metadata collection and analysis. The CDISC study data tabulation model (SDTM) and analysis dataset model (ADaM) standards are now required by the US FDA and Japanese Pharmaceuticals and Medical Devices Agency (PMDA), endorsed by the National Medical Products Administration (NMPA), and acknowledged by the EMA as the most proper standard for clinical trial data sharing and transfer.[34]

In collaboration with NCI, CDISC has also developed a glossary of CDE terms based on the EVS-controlled terminology following ISO/IEC 11179 standards[15] to facilitate digital exchange. CDISC can eventually be used to integrate Electronic Health Records (EHR) using Fast Healthcare Interoperable Resources (FHIR), and data mapping through the Biomedical Research Integrated Domain Group (BRIDG) model,[37] which will make it easier to harmonize data from clinical care and research and provide real-world data to the FDA to complement traditional clinical trials data. A few barriers exist to global CDISC use, including that many nonindustry trials do not adhere to CDISC standards (e.g., investigator-initiated research). Furthermore, as CDISC format is very different from what academic researchers prefer for data analyses, it may prove challenging to make CDISC a widely accepted format for PMD clinical research data community deposition and sharing. This may necessitate further communication and community consensus on how to either adapt current standards or develop tools to convert existing CDISC data sets to a new, mutually acceptable data standard. Any new tools developed for this purpose must be financially feasible and should be designed to optimize efficacy through user-friendly, automated processes. This new tool should also be "findable," deposited in publicly accessible databases such as "FAIRsharing.org" which collates metadata requirements for a range of communities (e.g., MIAME guidelines or CDISC Implementation guidelines to maximize community review, input, and refinement.

### 3.4. PMD Community Oversight

The PMD community is global but inherently limited in the numbers of molecularly confirmed patients interacting with the

growing array of stakeholders within the research and clinical care spectrum. Implementation GUID tags for all clinical research data being generated and shared is of critical importance to avoid data redundancy and misinterpretation and facilitates follow-up analyses from multiple resources collecting clinical research data on the same subject.[38,39] GUID tags require patient informed consent, but once assigned provide a path to link deidentified datasets from different sources at varying genetic, phenotypic, clinical trial, and multiomic levels into data repositories. GUID derivatives and deidentified data should be generated for specific data distribution and publication.[40]

The PMD community will need to decide on the semantic specifications of clinical research data to be captured and shared. For industry trials, ICD-10,[41] ICIMD,[14,42] and Orphacodes[43] are used as disease codes. Medications are usually coded and classified according to the World Health Organization (WHO) drug dictionary,[44] whereas adverse events for studies carried out after 2005 are usually coded in MedDRA[45] and older studies are coded in COSTART.[46] Some of these data dictionaries change regularly over time, leading to discrepancies in data coding that may prevent successful data aggregation. While deposited clinical research data would ideally be recoded, an estimated 10–20% would require manual coding that is time and resource intensive. Rare disease research requires granularity in phenotypic descriptions of subjects using defined ontologies, such as Human Phenotype Ontology (HPO)[47] terms, and knowing the exact circumstances of each data point to enable its accurate interpretation. For example, while timing from initial symptom onset to study entry may be a useful factor to consider, often neither exact nor approximate dates are collected; a patient with Leber Hereditary Optic Neuropathy (LHON) enrolled 4 weeks after visual loss symptom onset and one enrolled 4 years after symptom onset may have the same visual disability but quite different treatment response.

A new model of governance for data management, evaluation, and processing of data requests should be established for whichever community data aggregation model is selected. Sustainability and governance will ideally be shared by PMD professional societies, advocacy groups, and industry partners, and could potentially lead to a Disease Monitoring Program where Good Clinical Practice (GCP)-compliant datasets in standardized, regulated data formats would not only advance scientific and medical knowledge of PMD but would also serve as an orphan drugs product-specific registry to fulfill postmarketing regulatory requirements.[25]

## 4. Deposition Options for Large Datasets from PMD Clinical Research

Biomedical and scientific journals are an important conduit to clinical research data sharing and dissemination. The International Committee of Medical Journal Editors (ICMJE) adopted several guidelines and requirements since 2004 to enforce data sharing, increasing the transparency and validity of published clinical trial results.[48,49] This includes trial registration in an appropriate, publicly accessible database (i.e., ClinicalTrials.gov) prior to participants' enrollment, and commitment to share raw data at the time of publication. Key challenges remain including lack of oversight of the data shared by authors or clinical trial

sponsors and failure to share specific methodologies, rendering the validity of secondary analyses questionable. One option to address these challenges would be to require authors to include the details of their analytic data sets and their methodology when submitting data for publication, but this would require substantial oversight from the ICMJE and might inhibit some authors from publishing for fear of competition. A Creative Commons license is another approach to enable appropriate access to data by encouraging "data owners" to disclose analytical methods in reusable forms (e.g., a CWL or KNIME, snakemake or BioCompute Object) and to specify unambiguously the terms of use and ownership of the computational protocol which could potentially remedy that concern.

Existing data sharing platforms are also available that may be considered by the PMD clinical research community, including Yale Open Data Access project (YODA),[50] ClinicalStudyDataRequest.com,[51] and Vivli.[52] These systems vary in their level of user-friendly interfaces, suitability for rare and complex diseases such as PMD, and cost, but most require substantial IT support and management. The Critical Path Institute has launched Rare Disease Cures Accelerator-Data and Analytics platform (RDCA-DAP)[53] with backing from FDA and with plans to extend to EMA, which may offer a no-cost model to utilize as a starting point to standardize and integrate data sets from PMD clinical research studies.

Regardless of the platform used, a need exists for a PMD community repository that extends beyond the clinical trial data. Globally, several existing repositories could be leveraged for this purpose, including the Mitochondrial Disease Sequence Data Resource Consortium (MSeqDR), a global effort with Web interface established since 2014 to collect, curate, and share knowledge about PMD genes, variants, and deidentified genomic and phenotype data and tools for PMD analyses securely;[54] mitoSHARE,[55] the recently developed UMDF Patient-driven Registry as well as the Translational REsearch Advancing Therapy in MITOchondrial Diseases (TREAT MITO)[56] that is intended to function as a comprehensive PMD community repository for curation, governance, access oversight, sharing, distribution, and analysis of various data types, including health, genomic, and clinical trial data. PMD advocacy groups may optimally serve as central custodians to avoid institutional biases. In addition, GENOMIT[57] has now opened a global PMD registry, which includes multiple European countries and a site in Japan, and has established infrastructure for longitudinal, natural history and genomic data collection. Importantly, efforts to both leverage and integrate these existing platforms into a single, global repository will reduce competition between platforms, thereby minimizing redundancy and potential splintering of data across multiple registries (**Table 1**).

## 5. Conclusions

Improved harmonization and collaboration of global efforts in PMD therapeutic development will be important to move this rapidly growing field forward as effectively and efficiently as possible for the maximal benefit of both patients and other stakeholders. A unified, well-accepted system for clinical research structured data collection, sharing and deposition standards will accelerate trial design and outcomes. This paper represents

**Table 1.** Data Sharing and Integration for the PMD community and potential existing interoperability (HL-7, Health Level Seven International, CDISC, Clinical Data Interchange Standards Consortium; ICD, International Statistical Classification of Diseases and Related Health Problems; CPT, Current Procedural Terminology; LOINC, Logical Observations Identifiers, Names, Codes; REST: representational state transfer, API: application programming interface, EHR, electric health record, FHIR, fast healthcare interoperability resources; OHDSI, observational health data sciences and informatics; SDO, standards development organizations; SDTM, study data tabulation model; OMOP, observational medical outcomes partnership; HPO, human phenotype ontology).

| Potential data repository | Data interoperability |
| --- | --- |
| TREAT MITO[56] | Secure and permission-based data transfers between systems through web services, database-level integrations, file transfers, and other means, depending on the level of support provided by the other system. |
| | Standards supported by the platform include HL-7, CDISC, ClinicalTrials.gov and standard terminologies such as ICD, CPT, LOINC, and other data element standards published. |
| MSeqDR[54] | MSeqDR leverages the REST API extensively for data exchange with external resources and for interoperability. REST query will return in JSON format which is both machine- and human friendly. The genomic variant database MSeqDR-openCGA is implemented with the open-source OpenCB and OpenCGA packages, which also employ the REST interface, as well as command-line access and programming interfaces for Python and JAVA, among others. This will allow integration of external resources that provide REST API, EHR systems via FHIR. |
| MitoShare[55] | Standards development organizations (SDOs) such as HL7, CDISC, OHDSI, and others are focused in this space. The MitoSHARE platform provides a Service Bus API that allows for transformation from this operational JSON-Format into the various standards and models from the SDOs above such as FHIR, SDTM, OMOP, etc. |
| Genomit[57] | Data transformation and export via web-based APIs and SDTM. |
| | Uses standardized terminology such as HPO and ICD-10 to store clinical data. |

an important first step to unify stakeholders with the aim of recognizing and working toward standardizing expectations and structures for PMD clinical research data sharing. Future efforts will focus on developing metadata forms, protocols, and community accepted data sharing standards, as well as identifying interoperated platforms where data sharing and access will be housed. During this process, challenges pertaining to privacy legislation across countries, data security and identifiers, data access, oversight, and cost will have to be addressed.

## Supporting Information

Supporting Information is available from the Wiley Online Library or from the author.

## Acknowledgements

This work was funded in part by the Children's Hospital of Philadelphia Mitochondrial Medicine Frontier Program. The authors thank Xiaowu Gai, Ph.D., Priti and Jason Colquitt for contributing the technical operation systems for MSeqDR and MitoSHARE respectively. The authors are grateful for the encouragement and guidance of Myles Axton, Ph.D.

## Conflict of Interest

M.J.F. is scientific advisory board member with equity interest in RiboNova, Inc., and scientific board member as paid consultant with Khondrion and with Larimar Therapeutics; has previously been or is currently engaged with lifescience companies involved in mitochondrial disease therapeutic preclinical and/or clinical stage development as a paid consultant (Astellas (formerly Mitobridge) Pharma Inc., Autobahn Rx, Casma Therapeutics, Clarus Therapeutics, Cyclerion Therapeutics, Epirium Bio, HealthCap, Imel Therapeutics, Minovia Therapeutics, Abliva (formerly NeuroVive Pharmaceutical AB), Stealth BioTherapeutics, Zogenix, Inc.) and/or a sponsored research collaborator (AADI Bioscience, Astellas (formerly Mitobridge), Epirium Bio (formerly Cardero Therapeutics), Cyclerion Therapeutics, Epirium Bio, Imel Therapeutics, Khondrion, Minovia Therapeutics Inc., Mission Therapeutics, NeuroVive Pharmaceutical AB, PTC Bio, Raptor Therapeutics, REATA Inc., Reneo Therapeutics, RiboNova Inc., Standigm Inc., Stealth BioTherapeutics, and United Mitochondrial Disease Foundation); and receives royalties from Elsevier and educational speaking honorarium from PlatformQ and Agios Pharmaceuticals. A.K. received a research grantand consulting payments from Stealth BT, Sanofi Genzyme, and Takeda; a research grant from Protalix Biotherapeutics and REATA Pharmaceutical Inc; research grants from Astellas (formerly MitoBridge), Cyclerion Therapeutics, PTC Therapeutics, Idorsia; consulting payments from Astellas (formerly MitoBridge), Alexion, Lumleian, Homology, Akros, Abliva (formerly NeuroVive Pharmaceutical AB), Zogenix. A.K. is on the medical advisory board of MitoAction and on the scientific and medical advisory board of the United Mitochondrial Disease Foundation; the Coriell Institute is a founding and board member of the mitochondrial care network, and TREAT MITO, and President of the Mitochondrial Medicine Society. J.C.C. is an employee of Minovia Therapeutics. J.C. is awarded The Chair in Genomic Medicine and generously supported by The Royal Children's Hospital Foundation. B.H.C. received a research grant and consulting payments from Stealth BT; research grant support and consulting payments from Astellas (formerly MitoBridge), Reneo, Zogenix (formerly Modis); consulting payments from Abliva (formerly NeuroVive Pharmaceutical AB) and Neuroene, CoA Therapeutics; research grant support from PTC (formerly BioElectron Technologies); and is on the scientific and medical advisory board of the United Mitochondrial Disease Foundation; serves on the board of directors of the Child Neurology Foundation; an officer and president elect of the Child Neurology Society; a consultant for the American Academy of Neurology (AAN) to the American Medical Association CPT Panel, Chairman of the Advocacy Committee and board of directors of the AAN. S.R. is a paid scientific board member of Khondrion; received consulting payments from Access Infinity, BioMedical Insights, Epirium, Neurovive, Partners4Access, Pfizer and Stealth Biotherapeutics, and scientific advisory board payments from Pretzel, Taysha, and Modis Therapeutics, Inc. (a wholly owned subsidiary of Zogenix, Inc.); is the UK Chief Investigator for the PTC Biotherapeutics MIT-E trial; and is on the medical advisory boards of the Lily Foundation and the Freya Foundation. T.K. received research grants and consulting payments from Santhera Pharmaceuticals, Gensight Biologics, Stealth Biotherapeutics, Apopharma Inc. and Retrophin Inc.; received consulting payments from Chiesi GmbH and Pretzel Therapeutics. T.K.

acknowledges support by the German Federal Ministry of Education and Research (Bonn, Germany) through grants to the German Network for Mitochondrial Disorders (mitoNET, 01GM1906A) and to the E-Rare project GENOMIT (01GM1920B). T.K. is a member of the European Reference Network for Rare Neurological Diseases (ERN-RND) and of the European Reference Network for neuromuscular diseases (EURO-NMD), cofunded by the European Commission. R.M. receives funds from Wellcome, Medical Research Council UK, UMDF, the Lily Foundation and the Ryan Stanford Appeal. R.M. has acted in a consultant capacity for Taysha, Minovia, Omeicos, and Modis/Zogenix, withpayment for this consultancy work to Newcastle University, UK. A.V.S. has received research grants from the ALS Association, ALS Finding A Cure foundation, ALS Hope foundation, and the NIH/NIA and NIH/NINDS; is on the Board of the ALD Connect Research consortium. F.S. has received research grants from PTC Therapeutics, Stealth BT, Reata Pharmaceuticals, Entrada Therapeutics, Horizon Pharma, and the NIH (U54 NS078059-11); is on the Board of the Mitochondrial Medicine Society; and is an investigator for NAMDC.

## Author Contributions

A.K. and L.E.M. contributed equally to this work. Conceptualization: A.K. and M.J.F.; Project administration—A.K., L.E.M., and M.J.F.; Writing-original draft; A.K., L.E.M., J.C.C., M.H.C., J.C., B.H.C., Y.K., T.K., C.L., R.v.M., R.M., S.P., S.R., F.S., A.V.S., P.Y., M.J.F. Writing-review and editing: A.K., L.E.M., M.J.F. Approval of final version: A.K., L.E.M., J.C.C., J.C., B.H.C., T.K., Y.K., C.L., R.v.M., R.M., S.P., S.R., F.S., A.V.S., P.Y., M.J.F.

## Peer Review

The peer review history for this article is available in the Supporting Information for this article.

## Keywords

clinical trials, data sharing, FAIR standards, primary mitochondrial disease, therapeutic developments

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

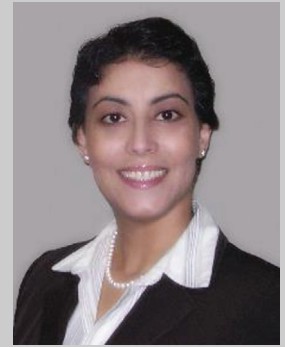

**Amel Karaa** is an internist and clinical geneticist by training and is the director of the mitochondrial disease program at the Massachusetts General Hospital in Boston. She received the 2013 United Mitochondrial Disease Foundation (UMDF) Fellowship. Dr. Karaa is overseeing clinical care, conducting clinical research and trials for mitochondrial disease patients. She is the current president of the Mitochondrial Medicine Society and sits on the scientific and medical board of many advocacy groups. Dr. Karaa is a founder and board member of the Mitochondrial Care Network (MCN) and the founder of TREAT MITO, a clinical research consortium for primary mitochondrial diseases.

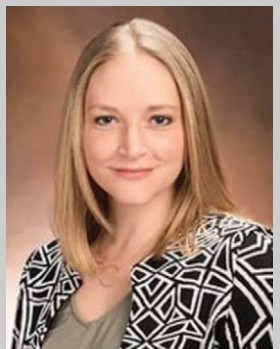

**Laura E. MacMullen** is a clinical research project manager in the Mitochondrial Medicine Frontier Program at Children's Hospital of Philadelphia (CHOP). Laura is a certified clinical research coordinator with a background in cognitive science, computational neuroscience, and behavioral health. Her focus at CHOP is to streamline processes for data integration and data analytics, design and implement natural history studies, manage start-up for clinical trials, and optimize research and clinical data management.

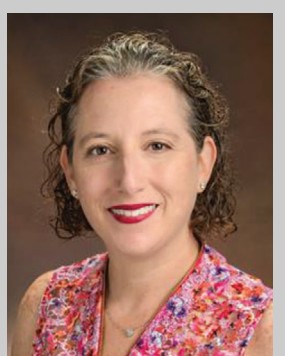

**Marni J. Falk** is a distinguished chair, professor, clinical geneticist, and executive director in the Mitochondrial Medicine Frontier Program (MMFP) in the Division of Human Genetics within the Department of Pediatrics at Children's Hospital of Philadelphia and University of Pennsylvania Perelman School of Medicine. She is the principal investigator of an active translational research laboratory group investigating the causes and global metabolic consequences of mitochondrial disease, and targeted therapies, in *Caenorhabditis elegans*, zebrafish, mouse, and human tissue models of respiratory chain dysfunction. Dr. Falk is also PI on clinical trials to better characterize and test candidate therapies in mitochondrial disease patients.

