## [**Supplementary Information**: Record of Transparent Peer Review · Advanced Genetics]

Community consensus guidelines to support FAIR data standards in clinical research studies in primary mitochondrial disease

Amel Karaa^{*}, Laura E. MacMullen, John C. Campbell, John Christodoulou, Bruce H. Cohen, Thomas Klopstock, Yasutoshi Koga, Costanza Lamperti, Rob van Maanen, Robert McFarland, Sumit Parikh, Shamima Rahman, Fernando Scaglia, Alexander V. Sherman, Philip Yeske, Marni J. Falk^{*}

*Corresponding

Review timeline: Date Submitted: 17 Sep 2021
 Editorial Decision: 22 Oct 2021
 Revision Received: 8 Nov 2021
 Accepted: 9 Nov 2021

Editor: Myles Axton

Initial Editorial Evaluation	17 Sep 2021
-------------

Summary

The community led by geneticists conducting mitochondrial disease clinical trials agree that there is rich data in small numbers and that certain fields including metadata for clinical phenotypes collected during these trials need to be shared more broadly so that measurements can be standardized and knowledge gained reused across trials. The data sharing arrangements and data schema will require more discussion, but the case for FAIR data solutions needs to be made now.

Scope

Do the research, methods or topics fit within the aims of this, or another journal?

FAIR data for rare, including mitochondrial, diseases is a core purpose for the journal. We agree that metadata sharing leads to standardization and reuse of phenotype and clinical data improving the treatment of genetic disorders, including mitochondrial diseases.

1 st Peer Review	20 Sep 2021-22 Oct 2021
-------------------------

Reviewer #1

Excellent topic and very much something that needs to be published on and discussed. The topic was very well articulated, documented, and covered.

Reviewer #2

===== Notes =====

Comments here are relative to the PDF page numbers, not the embedded page number.

This paper has a lot of important content and I am utterly convinced its goal is absolutely vital. While my major concerns are strong, I think it will be an important contribution if it is rewritten with a powerful vision of what it is trying to achieve, and a strong editor to make sure all the content supports that vision.

===== Major concerns =====

While this paper is critically important in its intent, it doesn't rise to the level of a convincing argument. It is a broad-brush consolidation of published and author-held ideas about the advantages and importance of a more open approach to data sharing, and the technical details that come into play. However it is not clear how the authors intend to win the argument with the uneven structure and content that the paper brings to an extremely complicated subject. Many critically important challenges and opportunities are only mentioned with a single sentence, without further discussion. Many words are expended on justifying the value of metadata and standards, but do not transition into a concrete plan for creating and adopting the necessary metadata and standards.

2.1 Either the paper could be rewritten as an opinion piece, that skips many of the technical details in favor of assertions about the importance of the interoperability mission to this data type. (This would in fact be "an important first step to unify stakeholders toward recognizing and working toward standardizing expectations and structures for PMD clinical research data sharing.")

2.2 Or it needs much deeper treatment of each section that currently exists as lists of existing approaches or technologies. It says it will "describe the current landscape of PMD clinical research data sharing, obstacles, and opportunities, including ways to encourage clinical research data sharing among diverse stakeholders."

2.3 It currently spends considerable time mentioning many (but not all) approaches, notionally some of the opportunities, and a few (not nearly enough) of the challenges, but in no case does it propose a particular approach, strategy, or tactic that can be taken to begin the process. To be specific, it is "A proposal ... to develop a unified system that securely enables clinical research structured data sharing and supports data deposition standards across clinical consortia and research groups to improve harmonization and collaboration of global efforts in PMD therapeutic development." only by declaration, not through any specific proposals.

===== Significant concerns =====

2.4 The terms 'metadata' and 'data' are used interchangeably, or at least in succession without clarification, and it is not precise about defining terms like schema, standards, formats, and templates.

For example:

p10 ln 15-19 "A common metadata sharing scheme would encompass an operating system that could evaluate and index the diverse variety of available data into an aggregated, pooled, global catalogue of data content." Metadata schemes do not provide the features this and following sentences attest to. This sentence describes 'a scheme to share data in a common way' -- not metadata, and not leaving ambiguity about what modifiers like 'common' are modifying (metadata or the scheme), or indeed even mean (frequently seen? commonly understood?)

2.5 Sections are unevenly addressed, for example:

p15 ln 53 By the end of this section, we have said nothing that is specific to large data sets, and have not said anything about Data Sharing Structures that appear in the title ("Comprehensive Data Sets and Data Sharing Structures for PMD clinical research data").

===== Detailed comments =====

2.6 p8 lines 6-12: Not clear whether the issue here is that the background knowledge gaps are a priori and the trials are burdensome in their attempts to collect the data, or the gaps occur in conduct of the trials and it's post-analysis or re-analysis/re-use that is trying to address the gaps. Or both.

2.7 p9 ln 52 "High-quality, granular data are also lacking, often with missing fields and data field errors that prevent accurate

comparisons across study protocol elements." What is it that has missing fields and data field errors? The high-quality granular data, or something else?

2.8 p10 In 31-38 "The PMD patient community would likely find greater motivation to participate in clinical research studies if they felt confident that all research subject data points would be maximally utilized to apply learnings to other PMD clinical trials, as well as to more common diseases involving secondary mitochondrial dysfunction." Insofar as this begins a new theme,

(a) perhaps start a new paragraph, and

(b) can you find a citation that supports it?

2.9 p12 In 22-24 "Typical requirements include data anonymization or coding..." Please make clear the meaning of 'coding' here (what is being coded in what way?). Assuming you are coding the data, disambiguate between encoding for security, and assigning a category or code.

2.10 p12 In 54 "The PMD community should seek to advance the discussions" What discussions?

2.11 p12-13 In 58,1 "These collaborative discussions are necessary to achieve the goal of community sharing in acceptable formats and data standards from PMD clinical research studies" I can't quite parse "in acceptable formats and data standards", one doesn't "share data in a data standard", one shares data *using* a data standard. (Reword.)

2.12 p13 In 45-50 "A first step in this process is to define and identify the type of data that need to be captured and standardized using structured data formats." In the metadata field, it is important to be relatively crisp when talking about both data (research information itself) and metadata (information about that research information), even while recognizing that metadata is a form of data. And also it's useful to be precise about using the word 'type'. Here I get confused by the switching back and forth of data (this paragraph) and metadata (next paragraph), and use of 'type of data that need to be captured' (floats vs ints? domain-specific vs cross-domain? data vs metadata? those are all types of data) when I think you mean 'data attributes that need to be captured'.

2.13 p13 In 54-56 "Metadata are deliberate, structured data that collate and analyze data from individual studies." Before collating and analyzing data, metadata is first used to *describe* that data. Also, not clear what 'deliberate' adds to this sentence.

2.14 p14 In 5-6 "community-approved schema (list of defined data points)". This phrasing doesn't make sense to me, "list of the metadata attributes that can and should be captured" would be in alignment. Some of those attributes may be a list of the variables in the data; that may also be in the "standards" list below, but a community schema can be a standard. See next item.

2.15 p14 In 5-8 "and standards (content, data value, and data structure)" Standards is too broad a term here. 'The community-approved schema may specify some or all of the 'standards'. I'm not sure what 'content' refers to here—typically it means what attributes are included in the data, and that would be appropriate for a *content standard*, which may be the appropriate term to use here. But there is typically just one for a given data set, so plural isn't a great choice given schema was singular.

2.16 p14 In 3-10 This sentence is good as far as it goes, I believe that agreement on semantic specifications (how to specify, e.g., using globally unique identifiers compatible with RDF) and semantic content (which vocabularies to use) are also required. Possibly they are embedded into either or both of the preceding key items. Note that emphasis on vocabularies in citation 25—that emphasis suggests that semantic specifications should be in this introductory section.

2.17 p14 In 10 This (second) citation 25 does not contain justification for the preceding sentence or phrase. Citation 26 does not appear to be it either.

2.18 p14 In 10-12 "For PMD, any and all properly consented, and privacy protected data collected in clinical research studies may be utilized." These things may be used for what, and how? We've started a new thread here and it was too big a jump. Generally you would not use privacy-protected data in the metadata, because you want the metadata to be generally accessible. Of course anything is possible with proper aggregation, anonymization, etc., but you need to capture that.

2.19 p14 In 10-30 Continuing the previous concern, the text is opaque about how all these things can be used and for what purpose. Even the explicit language "Demographics ... and adverse event data can be standardized to establish data dictionaries or broader the use of existing data standards like CDISC." We need concrete examples to go from the buzzwords to something real. ("standardizing" data can mean just about anything, and I can't tell what we're 'incorporating' the more granular data into.

2.20 p14 In 33-58 A long paragraph describes all the advantages of re-using and merging multiple large clinical datasets. Then one sentence (without attribution) describes a single concern about large clinical datasets, which doesn't seem to have anything to do with the topic of the paper. That sentence seems not useful.

2.21 p15 In 6-27 This paragraph almost exclusively describes the value of metadata for any data set, it is not unique to large data sets and we already explained the value of metadata in an earlier section, no?

2.22 p15 In 29-52 This paragraph has several ideas that deserve further discussions and exploration. The 'natural starting point' describes the *type of data* to start with, but skips over the more natural starting point of just sharing metadata. (A discussion of clinicaltrials.gov metadata is a good starting point for that topic, followed by what other metadata would be useful.) The order of challenges I would list (not just one) would be

(a) deciding what metadata beyond CT.gov to agree to share,

(b) figuring out how to get everyone on board with entering and sharing it,

(c) figuring out an appropriate amount of time to wait before sharing the metadata, and

(d) figuring out how to describe a data request and access protocol if the metadata indicates data of compelling value. (For starters.) The last 3 sentences simply pick a few of the many facets that are important, it isn't clear that list leads to improved understanding.

2.23 p15 In 56-61 Here again we are using data (in text) and metadata (in title and later in text) interchangeably. If we mean the same thing use the same word, if not clarify the difference.

2.24 p16 In 19-59 and p17 In 1-24 This is the most important content of the article so far. A discussion that ties together and compares these different approaches a bit more thoroughly would be welcome. A fourth approach, that of developing specifications that blend or surpass each of these, is not discussed but probably should be.

2.25 p17 In 51-53 "The PMD community will need to decide on the specifications of clinical research data to be captured and shared." Be more concrete and call these the terminology or semantic specifications.

2.26 p18 In 19-31 There is so much work embedded in this paragraph it is not adding much as it stands. It's a good example of where each line could be replaced by a paragraph of discussion, or the whole paragraph could be summarized in a single line.

Reviewer #3

We'd like to thank the authors for a well written manuscript of the `Perspective` article type.

The manuscript is organised around 5 sections, [1.introduction, 2.current landscape, 3. proposed formats and templates, 4. Deposition options, 5.Conclusions]

major comments:

3.1 * The authors nicely set the scene in the introduction section, laying out the key issues. However, as the reader progresses in the piece, one can not help but noticing a number of repetitions. The piece would therefore benefit from trimming where possible to make it more to the point and direct. (e.g. "the need to develop appropriate outcome measures" appears in section 1, but again in section 2 and 3)

3.2 * In section 2 (current landscape / Obstacles and opportunities), the authors provide a good overview of the challenges and

ongoing discussions, in particular when discussing the issue of 'standard data use requests'. Did the authors come across a resource such as the Data Use Ontology (DUO) and efforts by the Global Alliance for Genomics Health to establish passport (<https://www.ga4gh.org/news/ga4gh-passports-and-the-authorization-and-authentication-infrastructure/>). I was wondering if these efforts should be highlighted as on-going technical efforts to deal with the matter.

3.3 * in section 3 (proposed format and templates for sharing PMD data), the issue of 'templates' is introduced yet what 'templates' are isn't unambiguously defined. Are these sets of variables/CDE related to specific area and the presence of which can be ascertain by agents when receiving instance data from third parties/submitters?
If this is the meaning implied, would it be beneficial to insist on the deposition of such 'templates' in data standardisation efforts such as 'FAIRsharing.org', which collates metadata requirements for a range of communities (eg MIAME guidelines or CDISC Implementation guidelines).

3.4 Another comment is about the statement about using JSON-Format by MitoSHARE. While JSON-format is indeed well established and central to many a web service, it does not buy interoperability (no more than XML did 20 years ago). For instance FHIR can be serialised as JSON but it is distinct from MitoSHARE JSON message. The syntax while shared would not address the issue of (semantic) interoperability and minimal content availability for consumption by third parties.

3.5 Finally, it should be made explicit that it will be the templates that could make 'general purpose' models (such as CDISC) specific to the PMD and associated rare diseases. Hence, the importance of making any tentative templates 'findable' (e.g registration a CDISC implementation guide / FAIRsharing) to maximise community review, input and refinement.

3.6 * In section 4 (deposition options), could the authors clarify how "a creative commons license " would address the concern of sharing the details of their analytical data sets and their methodology'?
Here, it seems there are 2 issues. First, encouraging authors to disclose analytical methods in reusable forms (e.g. a CWL or KNIME or a snakemake workflow, even better a BioCompute Object since this is a standard endorsed by the US regulatory agency as far as I know)
Second, the issue of specifying unambiguously the terms of use and ownership of the computational protocol.

3.7 The section may also benefit from have a table showing interoperability potential between the various databases listed. For instance, which syntax standards are shared between TREAT MITO and GenoMIT or ClinicalTrials.gov, same for the semantics and controlled terminologies used.

3.8 Another aspect which would have been nice to consider is the machine actionability of the databases, in other words if standardised application programming interfaces are available (such as open API / smartAPI for REST services) and if not, why these would matter to enable FAIR data sharing for PMD.

3.9 * Figure 1:
might be worth trying to map existing/relevant syntactic and semantic data standards

3.10 * Conclusion:
It is somewhat a bit of missed opportunity not to point to a PMD specific web site / forum / collaborative develop framework to highlight the wealth of existing resources and calling for reviewing any 'embryonic' template for PMD research? Would that be a role for RDCA-DAP?

minor comments:

3m1 - page 15, lines 5-6: "Efforts ...has led to recommendations" -> "... have led ..."

Editor comments for author – seen by adjudicating reviewer #4 prior to sending decision	
Reviewer 1-3 comments	Editor recommendation
2.3 in no case does it propose a particular approach, strategy, or tactic that can be taken to begin the process. 3.5 templates that could make 'general purpose' models (such as CDISC) specific to the PMD and associated rare diseases. Hence, the importance of making any tentative templates 'findable' (e.g registration a CDISC	ED1 the central purpose of the Perspective is to declare the PMD partners are making their CDISC based metadata fields findable, licensing software in an explicit and machine-readable way (such as BioCompute Object with an appropriate license), declaring a set of acceptable access

implementation guide / FAIRsharing) to maximise community review, input and refinement. 3.6 BioCompute Object since this is a standard endorsed by the US regulatory agency as far as I know) Second, the issue of specifying unambiguously the terms of use and ownership of the computational protocol.	protocols for data (independent of the level of access declared). The actual fields and model that go beyond CDISC fields can be mocked up in a figure or mapped in a table to existing FAIRsharing entries.
2.1 Either the paper could be rewritten as an opinion piece, 2.2 Or it needs much deeper treatment .. of existing approaches or technologies. 2.25 p17 In 51-53 "The PMD community will need to decide on the specifications of clinical research data to be captured and shared." Be more concrete and call these the terminology or semantic specifications. 3.7 have a table showing interoperability potential between the various databases listed. For instance, which syntax standards are shared between TREAT MITO and GenoMIT or ClinicalTrials.gov, same for the semantics and controlled terminologies used. 3.9 * Figure 1: might be worth trying to map existing/relevant syntactic and semantic data standards	ED2 Declare the way forward (ED1 above) review, map and call for new semantic specifications in this Perspective using Fig 1 and a Table mapping existing standards. #4 says that it is imperative to consider the organizational aspects along with the technical and show the mapping in this article. The next article details the agreed metadata form and protocols, the third discusses the way in which the clinical data collection and phenotypes will be reported and interoperated
2.4 The terms 'metadata' and 'data' are used interchangeably, or at least in succession without clarification, and it is not precise about defining terms like schema, standards, formats, and templates.	ED3 Define what you are calling data (the clinical measurements, genotypes and secure information) and the metadata (accession codes, licenses, table headings, disease names, gene names, chemical entities, units etc.). Metadata entities come from ontologies, vocabularies and schemas. Data entries come from labs, clinics, patients and journals.
2.22 (d) figuring out how to describe a data request and access protocol 3.2 'standard data use requests'... ga4gh-passports 3.8 machine actionability of the databases, .. open API / smartAPI for REST services) 3.10 PMD specific web site / forum / collaborative develop framework	ED4 Suggest options for data requests to be handled by a community format like those templated by GA4GH and EGA and via a project specific website

Reviewer #4 – adjudicating reviewer

Comments on the Editors’ recommendations:

ED1: I agree with the Editors’ recommendation that the purpose of the Perspective is the authors declaration, and that specific solutions and implementations further down the process can be reported in future publications as mentioned in ED2.

- Defining the objective for this declaration is just as important (see minor comment 4.3 below).
- The declaration of this Perspective clearly show that the PMD community wants to make their clinical research data FAIR. This needs to be stated more clearly (see minor comment 4.4 and 4.5 below).
- A figure or FAIRsharing deposit of the “actual fields and model that go beyond CDISC fields” if available, otherwise include in the future publication.

ED2: I agree with the Editors’ recommendation to separate the different stages of this effort into more articles. However, next to the technical implementations the authors also need to consider the organizational aspects to reach its goals, and how they map to the different stages.

ED3 & ED4: Agree!

Minor comments:

4.1 Next to a collaborative PMD community effort, have the authors considered to which degree to align with efforts already taking place in other rare disease communities? Both to avoid reinventing existing solutions (save time and money), but also to ensure that the chosen solutions are interoperable across communities.

4.2 On page 7 line 45-47, “Data harmonization efforts across registries are underway in several rare disease groups in order to streamline therapeutic efforts.”, and on page 15 line 28/29-31, “Several other models of metadata sharing are being developed for rare disorders, and each platform has benefits and limitations.”. This needs references and more attention.

4.3 The authors mention several objectives for why data sharing can benefit the PMD community, one of this is sharing negative results, which amongst others would prevent duplications of experiments. These are strong points to describe to the stakeholders. Unfortunately, the objectives are shattered and sometimes repeated in the manuscript. Consider collecting these into a more focused section and ensure that these objectives lead the way to the solutions that you want to implement.

4.4 The term “FAIR (data) sharing” is used several times in the manuscript including the title, but what does it mean? Do the authors mean FAIR data (which goes beyond data sharing)? Or do they mean features from FAIR data that contribute to data sharing? Please consider changing this term or clearly define with references what is meant!

4.5 Are these data going to be made FAIR (or not)? This is not specifically stated in the paper, although FAIR appears in the title, in the keywords, and manuscript classification, but only mentioned twice in the manuscript without making any specific statement about it (page 10 line 14/15 and page 15 line 8).

4.6 On page 15 line 8 “... recommendations by FORCE11 to follow the FAIR principles ...”, the FORCE11 website is cited but not the original FAIR principles paper <https://doi.org/10.1038/sdata.2016.18>

4.7 On page 7 line 33, remove space before comma!

4.8 On page 13 line 51, ‘aide’ > ‘aid’.

1 st Editorial Decision	22 Oct 2021
-------------

Editorial decision: Revise addressing editor and reviewer recommendations

Editor’s understanding of the reviews

Reviewer #1 Recommends: Accept without revision

Reviewer #2 Recommends: Major revision

Reviewer #3 Recommends: Major revision

Reviewer #4 Recommends: Revise addressing editor and reviewer recommendations

Author’s Response to 1 st Review	8 Nov 2021
---	------------

These are the main reviewer recommendations that the editors believe will make the biggest improvement to this article. **Please do address all reviewer comments listed in the decision letter in your point-by-point response** (you may continue this table to do so if you wish). We hope this summary helps you to understand our decision and expedites the revision process. We value feedback from author and referees alike.
AdvGenet@wiley.com

Reviewer comments	Editor recommendation	Author reply	Changes to Manuscript
2.3 in no case does it propose a particular approach, strategy, or tactic that can be taken to begin the process. 3.5 templates that could make 'general purpose' models (such as CDISC) specific to the PMD and	ED1 the central purpose of the Perspective is to declare the PMD partners are making their CDISC based metadata fields findable, licensing software in an explicit and machine-readable way (such as BioCompute Object with an	We agree with the editor that this manuscript is the initial step for the PMD community to “declare” the need for FAIR data standards. We have updated the manuscript body and added text to clarify this point.	Changes were made throughout the manuscript (see track changes version). Table 1 can be found page 20.

associated rare diseases. Hence, the importance of making any tentative templates 'findable' (e.g registration a CDISC implementation guide / FAIRsharing) to maximise community review, input and refinement. 3.6 BioCompute Object since this is a standard endorsed by the US regulatory agency as far as I know) Second, the issue of specifying unambiguously the terms of use and ownership of the computational protocol.	appropriate license), declaring a set of acceptable access protocols for data (independent of the level of access declared). The actual fields and model that go beyond CDISC fields can be mocked up in a figure or mapped in a table to existing FAIRsharing entries.	We have also added a Table 1 to clarify the current Data type and format used in the current PMD platforms for data collection.	
2.1 Either the paper could be rewritten as an opinion piece, 2.2 Or it needs much deeper treatment .. of existing approaches or technologies. 2.25 p17 In 51-53 "The PMD community will need to decide on the specifications of clinical research data to be captured and shared." Be more concrete and call these the terminology or semantic specifications. 3.7 have a table showing interoperability potential between the various databases listed. For instance, which syntax standards are shared between TREAT MITO and GenoMIT or ClinicalTrials.gov, same for the semantics and controlled terminologies used. 3.9 * Figure 1: might be worth trying to map existing/relevant syntactic and semantic data standards	ED2 Declare the way forward (ED1 above) review, map and call for new semantic specifications in this Perspective using Fig 1 and a Table mapping existing standards. #4 says that it is imperative to consider the organizational aspects along with the technical and show the mapping in this article. The next article details the agreed metadata form and protocols, the third discusses the way in which the clinical data collection and phenotypes will be reported and interoperated	We again agree with the editor that this manuscript is the initial step for the PMD community to "declare" the need for FAIR data standards. We have added a paragraph in both the introduction and conclusion to reflect this more clearly.	Paragraph was added in the introduction on page 8. Paragraph added in the conclusion on page 22.
2.4 The terms 'metadata' and 'data' are used interchangeably, or at least in succession without clarification, and it is not precise about defining terms like schema, standards, formats, and templates.	ED3 Define what you are calling data (the clinical measurements, genotypes and secure information) and the metadata (accession codes, licenses, table headings, disease names,	Thank you for pointing out the need for this clarification. When these terms are used in succession it is to indicate sharing of both data and metadata, and while metadata is defined in a later section we	Added definitions as suggested of data and meta data to the introduction. Reviewed manuscript to ensure proper use of terms.

	gene names, chemical entities, units etc.). Metadata entities come from ontologies, vocabularies and schemas. Data entries come from labs, clinics, patients and journals.	believe that including the definitions early in the manuscript provides the necessary clarification	
2.22 (d) figuring out how to describe a data request and access protocol 3.2 'standard data use requests'... ga4gh-passports 3.8 machine actionability of the databases, ... open API / smartAPI for REST services) 3.10 PMD specific web site / forum / collaborative develop framework	ED4 Suggest options for data requests to be handled by a community format like those templated by GA4GH and EGA and via a project specific website	Thank you for this comment. We have rectified this in the body of the manuscript.	Included a discussion on GA4GH efforts in section 2.

Reviewer #1

Excellent topic and very much something that needs to be published on and discussed. The topic was very well articulated, documented, and covered.

We thank the reviewer for these comments.

Reviewer #2

===== Notes =====

Comments here are relative to the PDF page numbers, not the embedded page number.

This paper has a lot of important content and I am utterly convinced its goal is absolutely vital. While my major concerns are strong, I think it will be an important contribution if it is rewritten with a powerful vision of what it is trying to achieve, and a strong editor to make sure all the content supports that vision.

===== Major concerns =====

While this paper is critically important in its intent, it doesn't rise to the level of a convincing argument. It is a broad-brush consolidation of published and author-held ideas about the advantages and importance of a more open approach to data sharing, and the technical details that come into play. However it is not clear how the authors intend to win the argument with the uneven structure and content that the paper brings to an extremely complicated subject. Many critically important challenges and opportunities are only mentioned with a single sentence, without further discussion. Many words are expended on justifying the value of metadata and standards, but do not transition into a concrete plan for creating and adopting the necessary metadata and standards.

2.1 Either the paper could be rewritten as an opinion piece, that skips many of the technical details in favor of assertions about the importance of the interoperability mission to this data type. (This would in fact be "an important first step to unify stakeholders toward recognizing and working toward standardizing expectations and structures for PMD clinical research data sharing.")

→ We have incorporated many of the changes suggested within this manuscript and added a paragraph in the conclusion to detail the next important steps for the PMD community to achieve FAIR data standards. We hope that these changes have clarified the message of the paper.

2.2 Or it needs much deeper treatment of each section that currently exists as lists of existing approaches or technologies. It says it will "describe the current landscape of PMD clinical research data sharing, obstacles, and opportunities, including ways to encourage clinical research data sharing among diverse stakeholders."

→ A table (Table 1) has been added detailing the technical aspects of existing PMD community infrastructures and how they might share data.

2.3 It currently spends considerable time mentioning many (but not all) approaches, notionally some of the opportunities, and a few (not nearly enough) of the challenges, but in no case does it propose a particular approach, strategy, or tactic that can be taken to begin the process. To be specific, it is "A proposal ... to develop a unified system that securely enables clinical research structured data sharing and supports data deposition standards across clinical consortia and research groups to improve harmonization and collaboration of global efforts in PMD therapeutic development." only by declaration, not through any specific proposals.

→ Yes, we agree that the intent of this first publication is to "declare" the PMD community's understanding of the gaps and the need to address them. Future publications will detail the specific approaches to support FAIR Data Standards and sharing.

===== Significant concerns =====

2.4 The terms 'metadata' and 'data' are used interchangeably, or at least in succession without clarification, and it is not precise about defining terms like schema, standards, formats, and templates.

Thank you for pointing out the need for this clarification. When these terms are used in succession it is to indicate sharing of both data and metadata, and while metadata is defined in a later section we believe that including the definitions early in the manuscript provides the necessary clarification. We have added definitions as suggested of data and meta data to the introduction. Reviewed manuscript to ensure proper use of terms.

For example:

p10 In 15-19 "A common metadata sharing scheme would encompass an operating system that could evaluate and index the diverse variety of available data into an aggregated, pooled, global catalogue of data content." Metadata schemes do not provide the features this and following sentences attest to. This sentence describes 'a scheme to share data in a common way' -- not metadata, and not leaving ambiguity about what modifiers like 'common' are modifying (metadata or the scheme), or indeed even mean (frequently seen? commonly understood?)

This section has been updated as suggested using the appropriate terms and to provide additional clarification.

2.5 Sections are unevenly addressed, for example:

p15 In 53 By the end of this section, we have said nothing that is specific to large data sets, and have not said anything about Data Sharing Structures that appear in the title ("Comprehensive Data Sets and Data Sharing Structures for PMD clinical research data").

The point being made in this section is not that these features are specific to large data sets, but that large data sets and collaborative efforts to share this data are particularly crucial in *rare disease* research since typically key stakeholders only have access to small data sets. We have edited the section header to more accurately summarize this section.

===== Detailed comments =====

2.6 p8 lines 6-12: Not clear whether the issue here is that the background knowledge gaps are a priori and the trials are burdensome in their attempts to collect the data, or the gaps occur in conduct of the trials and it's post-analysis or re-analysis/re-use that is trying to address the gaps. Or both. The background knowledge gaps are a priori and the burden of clinical trial data collection could be reduced if clinical trial design was informed by robust longitudinal data. We added clarifying language to indicate data collection as the source of clinical trial burden discussed here.

2.7 p9 In 52 "High-quality, granular data are also lacking, often with missing fields and data field errors that prevent accurate comparisons across study protocol elements." What is it that has missing fields and data field errors? The high-quality granular data, or something else?

Thank you for pointing out this error in sentence structure. We have edited this to indicate that it is the current data sets that have missing fields, highlighting the lack of high-quality data.

2.8 p10 ln 31-38 "The PMD patient community would likely find greater motivation to participate in clinical research studies if they felt confident that all research subject data points would be maximally utilized to apply learnings to other PMD clinical trials, as well as to more common diseases involving secondary mitochondrial dysfunction." Insofar as this begins a new theme,

(a) perhaps start a new paragraph, and

(b) can you find a citation that supports it?

We edited the manuscript to begin a new paragraph here and added supporting text with citation.

2.9 p12 ln 22-24 "Typical requirements include data anonymization or coding..." Please make clear the meaning of 'coding' here (what is being coded in what way?). Assuming you are coding the data, disambiguate between encoding for security, and assigning a category or code.

Added clarifying text.

2.10 p12 ln 54 "The PMD community should seek to advance the discussions" What discussions?

Added clarifying text: to advance discussions on overcoming challenges to data sharing.

2.11 p12-13 ln 58,1 "These collaborative discussions are necessary to achieve the goal of community sharing in acceptable formats and data standards from PMD clinical research studies" I can't quite parse "in acceptable formats and data standards", one doesn't "share data in a data standard", one shares data *using* a data standard. (Reword.)

Reworded.

2.12 p13 ln 45-50 "A first step in this process is to define and identify the type of data that need to be captured and standardized using structured data formats." In the metadata field, it is important to be relatively crisp when talking about both data (research information itself) and metadata (information about that research information), even while recognizing that metadata is a form of data. And also it's useful to be precise about using the word 'type'. Here I get confused by the switching back and forth of data (this paragraph) and metadata (next paragraph), and use of 'type of data that need to be captured' (floats vs ints? domain-specific vs cross-domain? data vs metadata? those are all types of data) when I think you mean 'data attributes that need to be captured'.

We have edited these sections as suggested.

2.13 p13 ln 54-56 "Metadata are deliberate, structured data that collate and analyze data from individual studies." Before collating and analyzing data, metadata is first used to *describe* that data. Also, not clear what 'deliberate' adds to this sentence.

We have edited this sentence as suggested.

2.14 p14 ln 5-6 "community-approved schema (list of defined data points)". This phrasing doesn't make sense to me, "list of the metadata attributes that can and should be captured" would be in alignment. Some of those attributes may be a list of the variables in the data; that may also be in the "standards" list below, but a community schema can be a standard. See next item.

We have edited this sentence as suggested.

2.15 p14 ln 5-8 "and standards (content, data value, and data structure)" Standards is too broad a term here. 'The community-approved schema may specify some or all of the 'standards'. I'm not sure what 'content' refers to here—typically it means what attributes are included in the data, and that would be appropriate for a *content standard*, which may be the appropriate term to use here. But there is typically just one for a given data set, so plural isn't a great choice given schema was singular.

We have edited as suggested.

2.16 p14 ln 3-10 This sentence is good as far as it goes, I believe that agreement on semantic specifications (how to specify, e.g., using globally unique identifiers compatible with RDF) and semantic content (which vocabularies to use) are also required. Possibly they are embedded into either or both of the preceding key items. Note that emphasis on vocabularies in citation 25—that emphasis suggest that semantic specifications should be in this introductory section.

We have changed the language in this section as suggested.

2.17 p14 ln 10 This (second) citation 25 does not contain justification for the preceding sentence or phrase. Citation 26 does not appear to be it either.

We thank the reviewer for catching this discrepancy, we have corrected the citation.

2.18 p14 ln 10-12 "For PMD, any and all properly consented, and privacy protected data collected in clinical research studies may be utilized." These things may be used for what, and how? We've started a new thread here and it was too big a jump. Generally you would not use privacy-protected data in the metadata, because you want the metadata to be generally accessible. Of course anything is possible with proper aggregation, anonymization, etc., but you need to capture that.

We have changed the sentence to clarify the message.

2.19 p14 ln 10-30 Continuing the previous concern, the text is opaque about how all these things can be used and for what purpose. Even the explicit language "Demographics ... and adverse event data can be standardized to establish data dictionaries or broader the use of existing data standards like CDISC." We need concrete examples to go from the buzzwords to something real. ("standardizing" data can mean just about anything, and I can't tell what we're 'incorporating' the more granular data into.

We added a sentence to explain the usefulness of this approach.

2.20 p14 ln 33-58 A long paragraph describes all the advantages of re-using and merging multiple large clinical datasets. Then one sentence (without attribution) describes a single concern about large clinical datasets, which doesn't seem to have anything to do with the topic of the paper. That sentence seems not useful.

While this is an important point in consideration of clinical trial design, the authors agree that it does not follow the thread here and we have deleted this sentence.

2.21 p15 ln 6-27 This paragraph almost exclusively describes the value of metadata for any data set, it is not unique to large data sets and we already explained the value of metadata in an earlier section, no? See previous comment in response to 2.5.

2.22 p15 ln 29-52 This paragraph has several ideas that deserve further discussions and exploration. The 'natural starting point' describes the *type of data* to start with, but skips over the more natural starting point of just sharing metadata. (A discussion of clinicaltrials.gov metadata is a good starting point for that topic, followed by what other metadata would be useful.) The order of challenges I would list (not just one) would be

(a) deciding what metadata beyond CT.gov to agree to share,

(b) figuring out how to get everyone on board with entering and sharing it,

(c) figuring out an appropriate amount of time to wait before sharing the metadata, and

(d) figuring out how to describe a data request and access protocol if the metadata indicates data of compelling value. (For starters.) The last 3 sentences simply pick a few of the many facets that are important, it isn't clear that list leads to improved understanding.

We thank the reviewer for these suggestions. These challenges were incorporated within the manuscript paragraph.

2.23 p15 ln 56-61 Here again we are using data (in text) and metadata (in title and later in text) interchangeably. If we mean the same thing use the same word, if not clarify the difference.

Edited as suggested.

2.24 p16 ln 19-59 and p17 ln 1-24 This is the most important content of the article so far. A discussion that ties together and compares these different approaches a bit more thoroughly would be welcome. A fourth approach, that of developing specifications that blend or surpass each of these, is not discussed but probably should be.

Table 1 has been added detailing the different approaches currently in use within the PMD community and their potential interoperability.

2.25 p17 ln 51-53 "The PMD community will need to decide on the specifications of clinical research data to be captured and shared." Be more concrete and call these the terminology or semantic specifications.

Edited as suggested.

2.26 p18 ln 19-31 There is so much work embedded in this paragraph it is not adding much as it stands. It's a good example of where each line could be replaced by a paragraph of discussion, or the whole paragraph could be summarized in a single line. The detailed protocols and technological interfaces to be proposed for the PMD community will be the subject of a subsequent publication. The authors feel that it would be outside of the scope of this initial "declaration" to include these details.

Reviewer #3

We'd like to thank the authors for a well written manuscript of the `Perspective` article type.

The manuscript is organised around 5 sections, [1.introduction, 2.current landscape, 3. proposed formats and templates, 4. Deposition options, 5.Conclusions]

major comments:

3.1 * The authors nicely set the scene in the introduction section, laying out the key issues. However, as the reader progresses in the piece, one can not help but noticing a number of repetitions. The piece would therefore benefit from trimming where possible to make it more to the point and direct. (e.g. "the need to develop appropriate outcome measures" appears in section 1, but again in section 2 and 3)

Thank you for your feedback. We have reviewed and cut down on repetition where appropriate. However, we believe section 3 makes a different point: specifically, that analysis of large clinical data sets helps support the need addressed in section 1.

3.2 * In section 2 (current landscape / Obstacles and opportunities), the authors provide a good overview of the challenges and ongoing discussions, in particular when discussing the issue of 'standard data use requests'. Did the authors come across a resource such as the Data Use Ontology (DUO) and efforts by the Global Alliance for Genomics Health to establish passport (<https://www.ga4gh.org/news/ga4gh-passports-and-the-authorization-and-authentication-infrastructure/>). I was wondering if these efforts should be highlighted as on-going technical efforts to deal with the matter.

We have incorporated these efforts into this section.

3.3 * in section 3 (proposed format and templates for sharing PMD data), the issue of 'templates' is introduced yet what 'templates' are isn't unambiguously defined. Are these sets of variables/CDE related to specific area and the presence of which can be ascertain by agents when receiving instance data from third parties/submitters?

If this is the meaning implied, would it be beneficial to insist on the deposition of such 'templates' in data standardization efforts such as 'FAIRsharing.org', which collates metadata requirements for a range of communities (eg MIAME guidelines or CDISC Implementation guidelines).

We have replaced "templates" with methods as the meaning here is to develop the "methodology" for data sharing.

Developing the “templates” that will be used for data capture and sharing through these methodologies, would be a subject of a subsequent community effort.

3.4 Another comment is about the statement about using JSON-Format by MitoSHARE. While JSON-format is indeed well established and central to many a web service, it does not buy interoperability (no more than XML did 20 years ago). For instance FHIR can be serialised as JSON but it is distinct from MitoSHARE JSON message. The syntax while shared would not address the issue of (semantic) interoperability and minimal content availability for consumption by third parties.

Details of MitoSHARE operations were added to Table 1 in an effort to address this comment.

3.5 Finally, it should be made explicit that it will be the templates that could make 'general purpose' models (such as CDISC) specific to the PMD and associated rare diseases. Hence, the importance of making any tentative templates 'findable' (e.g registration a CDISC implementation guide / FAIRsharing) to maximize community review, input and refinement.

We have added these ideas to the body of the manuscript on page 17.

3.6 * In section 4 (deposition options), could the authors clarify how "a creative commons license " would address the concern of sharing the details of their analytical data sets and their methodology'?

Here, it seems there are 2 issues. First, encouraging authors to disclose analytical methods in reusable forms (e.g. a CWL or KNIME or a snakemake workflow, even better a BioCompute Object since this is a standard endorsed by the US regulatory agency as far as I know)

Second, the issue of specifying unambiguously the terms of use and ownership of the computational protocol.

We have added these points to the manuscript.

3.7 The section may also benefit from have a table showing interoperability potential between the various databases listed. For instance, which syntax standards are shared between TREAT MITO and GenoMIT or ClinicalTrials.gov, same for the semantics and controlled terminologies used.

Table 1 was added with information about TREAT MITO and GenoMIT operational systems.

3.8 Another aspect which would have been nice to consider is the machine actionability of the databases, in other words if standardized application programming interfaces are available (such as open API / smartAPI for REST services) and if not, why these would matter to enable FAIR data sharing for PMD.

We added some of these details in Table 1 to address the reviewee's concerns

3.9 * Figure 1:

might be worth trying to map existing/relevant syntactic and semantic data standards

We have tried to incorporate some the existing data standard used in Table 1. A full “existing/relevant syntactic data standards” is outside the scope of this initial paper and will be proposed for the second publication.

3.10 * Conclusion:

It is somewhat a bit of missed opportunity not to point to a PMD specific web site / forum / collaborative develop framework to highlight the wealth of existing resources and calling for reviewing any 'embryonic' template for PMD research? Would that be a role for RDCA-DAP?

→ Concern addressed. We have added some updates on the PMD initiative

minor comments:

3m1 - page 15, lines 5-6: "Efforts ...has led to recommendations" -> "... have led ..."

Corrected

Reviewer #4 – adjudicating reviewer

Comments on the Editors' recommendations:

ED1: I agree with the Editors' recommendation that the purpose of the Perspective is the authors declaration, and that specific solutions and implementations further down the process can be reported in future publications as mentioned in ED2.

- Defining the objective for this declaration is just as important (see minor comment 4.3 below).

- The declaration of this Perspective clearly show that the PMD community wants to make their clinical research data FAIR. This needs to be stated more clearly (see minor comment 4.4 and 4.5 below).

- A figure or FAIRsharing deposit of the "actual fields and model that go beyond CDISC fields" if available, otherwise include in the future publication.

As stated above, we agree with the editor that this manuscript is the initial step for the PMD community to "declare" the need for FAIR data standards. We have added a paragraph in both the introduction and conclusion to reflect this more clearly.

ED2: I agree with the Editors' recommendation to separate the different stages of this effort into more articles. However, next to the technical implementations the authors also need to consider the organizational aspects to reach its goals, and how they map to the different stages.

We have added a paragraph in the conclusion to further delineate these future plans.

ED3 & ED4: Agree!

Minor comments:

4.1 Next to a collaborative PMD community effort, have the authors considered to which degree to align with efforts already taking place in other rare disease communities? Both to avoid reinventing existing solutions (save time and money), but also to ensure that the chosen solutions are interoperable across communities.

Yes, we agree with the reviewer. We have addressed similar efforts in other rare disease communities on page 12 (second paragraph).

4.2 On page 7 line 45-47, "Data harmonization efforts across registries are underway in several rare disease groups in order to streamline therapeutic efforts.", and on page 15 line 28/29-31, "Several other models of metadata sharing are being developed for rare disorders, and each platform has benefits and limitations.". This needs references and more attention.

References and examples are provided in the second to last paragraph of section 2 (current landscape).

4.3 The authors mention several objectives for why data sharing can benefit the PMD community, one of this is sharing negative results, which amongst others would prevent duplications of experiments. These are strong points to describe to the stakeholders. Unfortunately, the objectives are shattered and sometimes repeated in the manuscript. Consider collecting these into a more focused section and ensure that these objectives lead the way to the solutions that you want to implement.

We thank the reviewer for pointing this out. We have revised the manuscript whenever necessary to make these points clearer.

4.4 The term "FAIR (data) sharing" is used several times in the manuscript including the title, but what does it mean? Do the authors mean FAIR data (which goes beyond data sharing)? Or do they mean features from FAIR data that contribute to data sharing? Please consider changing this term or clearly define with references what is meant!

The term "FAIR" pertains to making data FAIR, ergo available for sharing. We have updated this notion across the manuscript to further clarify.

4.5 Are these data going to be made FAIR (or not)? This is not specifically stated in the paper, although FAIR appears in the title, in the keywords, and manuscript classification, but only mentioned twice in the manuscript without making any specific statement about it (page 10 line 14/15 and page 15 line 8).

We have updated the manuscript to include a description of and reference to FAIR data as a precursor for data sharing.

4.6 On page 15 line 8 "... recommendations by FORCE11 to follow the FAIR principles ...", the FORCE11 website is cited but not the original FAIR principles paper <https://doi.org/10.1038/sdata.2016.18>.

Added as requested

4.7 On page 7 line 33, remove space before comma!

Corrected.

4.8 On page 13 line 51, 'aide' > 'aid'.

Corrected.

2nd Editorial Decision

9 Nov 2021

The manuscript has been revised incorporating the reviewers' and editor's suggestions and we have now decided to accept the revised manuscript in principle, subject to the attached formatting requirements.